



Earth **Surface**
Dynamics

# Large wood as a confounding factor in interpreting the width of spring-fed streams

**Dana Ariel Lapides and Michael Manga**

Department of Earth and Planetary Science, University of California, Berkeley, CA 94720-4767, USA

**Correspondence:** Dana Ariel Lapides (danalapides@gmail.com)

**Abstract.** Spring-fed streams throughout volcanic regions of the western United States exhibit larger widths than runoff-fed streams with similar discharge. Due to the distinctive damped hydrograph of spring-fed streams (as compared to large peaks visible in the hydrographs of runoff-fed streams), large wood is less mobile in spring-fed than runoff-fed stream channels, so wood is more likely to remain in place than form logjams as in runoff-fed streams. The consequent long residence time of wood in spring-fed streams allows wood to potentially have long-term impacts on channel morphology. We used high-resolution satellite imagery in combination with discharge and climate data from published reports and publicly available databases to investigate the relationship between discharge, wood length, and channel width in 38 spring-fed and 20 runoff-fed streams, additionally responding to a call for increased use of remote sensing to study wood dynamics and daylighting previously unpublished data. We identified an order of magnitude more logjams than single logs per unit length present in runoff-fed streams as compared to spring-fed streams. Histograms of log orientation in spring-fed streams additionally confirmed that single logs are immobile in the channel so that the impact of single logs on channel morphology could be pronounced in spring-fed streams. Based on these observed differences, we hypothesized that there should be a difference in channel morphology. We found that spring-fed streams in our study are about 2 times wider than runoff-fed streams with similar mean discharge. A model for stream width in spring-fed streams based solely on length of wood is a better model than one derived from discharge or including both discharge and wood length. This study provides insights into controls on stream width in spring-fed streams.

## 1 Introduction

Leopold and Maddock (1953) first proposed a set of power laws to describe channel morphology based on discharge. Subsequent studies confirmed the existence of a relationship between discharge and width (e.g., Ferguson, 1986; Ackers, 1964; Stall and Fok, 1968), but the scatter in the relationship is large. There is a wealth of empirical correlations to describe width based on environmental conditions; however, the best relationships exhibit limited capacity to describe real channels (Gleason, 2015).

In certain cases, though, it may be possible to predict channel width more precisely. One example is that of spring-dominated or spring-fed streams. Spring-fed streams receive the bulk of their discharge from groundwater sources and thus exhibit relatively stable hydrographs (e.g., Whiting and

Moog, 2001; Manga, 1996). Compared to runoff-fed streams, spring-fed streams transport a proportionally larger amount of sediment in everyday flows than high-flow events, leading to different channel responses to disturbance, such as flow obstacles (Whiting and Stamm, 1995). Spring-fed streams are a promising test group for understanding some of the controls on stream width since their stable hydrographs reduce the number of variables impacting the channel.

Previous studies have identified differences between runoff- and spring-fed channels (e.g., Whiting and Moog, 2001; Griffiths et al., 2008). Whiting and Moog (2001) studied streams in the western US, primarily in the Oregon Cascades, and found that the spring-fed streams in their study ($0.005$–$8\,\mathrm{m^3\,s^{-1}}$) are significantly wider than their runoff-fed counterparts. Conversely, a study comparing spring-fed

to runoff-fed streams in Arizona ($10^{-3}$ m$^3$ s$^{-1}$) found that spring-fed streams exhibit lower width-to-depth ratios than runoff-fed streams (Griffiths et al., 2008). The streams studied by Whiting and Moog (2001) and Griffiths et al. (2008) are comparable in every aspect, save discharge and the presence of large wood (LW). The streams studied by Whiting and Moog (2001) had high discharge and significant amounts of LW, while the streams studied by Griffiths et al. (2008) had very low discharge and essentially no LW.

In many settings, including those considered in this study, LW is typically recruited through wind storms, death by bark beetles, and undercutting banks. The presence of LW increases variance in channel width, demonstrating the capacity to either constrict or widen (Montgomery et al., 2003). Channel widening associated with LW was observed by Trotter (1990), Nakamura and Swanson (1993), Hart (2002), and Faustini and Jones (2003), for example. Manga and Kirchner (2000) found that the presence of wood increases mean water depth, implying lower mean velocities but local velocity increases. Zhang et al. (2016) demonstrated that single logs can increase bank erosion via those local velocity increases, providing a mechanism for channel widening with the presence of LW. However, with multiple single logs in a stream, the effect is enhanced when single logs are very close together but dampened when they are moderately closely spaced (Zhang, Rutherfurd, and Marren, 2019). In contrast, removal of LW has been observed to cause rapid changes to channel form, including rapid channel widening (Bilby, 1984; Smith et al., 1993; Brooks and Brierley, 2000). The mechanism for LW constriction of channel width is stream bank stabilization by LW (Montgomery et al., 2003).

Despite evidence that LW impacts channel dimensions, LW was absent from early discussions of channel geometry (Gleason, 2015). We hypothesize that LW widens spring-fed streams. In general, the stability of LW in channels is related to flow characteristics of the stream and the size of LW (Bilby, 1984; Bilby and Ward, 1989; Berg et al., 1998; Gleason, 2015). Notably, Senter et al. (2017) showed that peak annual discharge has a large impact on LW mobility, and generally, hydrology is a good predictor of wood mobility (Kramer and Wohl, 2016). Thus, due to differing hydrograph behavior, peak events in runoff-fed streams may be able to mobilize wood, whereas the more stable hydrographs of spring-fed streams generally lie below the threshold for wood mobility, making LW more likely to be immobile in spring-fed but not runoff-fed streams. In order to assess this hypothesis, Hygelund (2002) measured orientations and diameters of wood in Oregon streams to determine whether wood was oriented with respect to the thalweg. They found that wood in runoff-fed channels was generally more oriented with flow, demonstrating mobility, and wood in spring-fed channels was generally aligned randomly or more perpendicular with flow, implying immobility.

We hypothesize that mobility promotes the development of logjams in runoff-fed streams (e.g., Martin and Benda,

2001) and explains the paucity of logjams in spring-fed streams, where single logs may dominate the population of LW. In addition to the impacts on channel widening, the presence of logjams may impact morphology by forcing a multi-threaded rather than a single-thread channel (Wohl, 2014; Polvi and Wohl, 2013). With a low abundance of logjams in spring-fed streams, we thus expect that the wood interaction mechanism explored by Zhang et al. (2016) for single logs in single-thread streams (i.e., an increase in bank erosion) may dominate, leading to channel widening associated with the presence of LW. With sufficient logs immobile in a channel, the consequent bank erosion would increase the reach-averaged width-to-depth ratio. In contrast, logjams may produce more variable effects on channel morphology or locally stabilize banks, causing channel constriction.

The purpose of this study is to examine the empirical relationship between LW and the morphology of spring-fed streams in order to identify statistically significant relationships. We also respond to a recent call by Kramer and Wohl (2016) to employ remote sensing to study wood dynamics and to daylight unpublished data on wood dynamics. Specifically, we investigated (1) wood orientation and frequency of logjams, (2) discharge and width of stream channels, and (3) length of LW and width of stream channels.

## 2 Field area

In this study, we work with 36 spring-fed streams and 20 runoff-fed streams across the western United States in the Oregon Cascades, southwestern Montana, eastern Idaho, northern Arizona, northern California, and the Ozarks in Missouri, and two additional spring-fed streams in El Tatio geyser field in Chile (Table 1). Bankfull discharge ranges from the approximately $10^{-3}$ m$^3$ s$^{-1}$ discharge springs in Arizona (Griffiths et al., 2008) to Big Springs, MO, at 13 m$^3$ s$^{-1}$ (USGS, 2018), with precipitation varying by only a factor of 4 in the North American examples. The streambeds generally consist of glacial outwash or alluvium. All streams included in this study have erodible banks. Streams in this study are generally single threaded with some examples of multi-threaded reaches in channels, generally coinciding with large amounts of LW.

The streams located in eastern Idaho and southwestern Montana are located in the easternmost part of the Columbia Plateau (Snake River Plain) and neighboring Middle Rocky Mountains physiographic provinces (Fenneman, 1931). The annual precipitation is 300–600 mm, with about 150 mm snowfall (Arguez et al., 2010). Mean annual temperatures range from 1 to 9 °C (Arguez et al., 2010). The area is underlain by Quaternary rhyolite and basalt (Christiansen and Blank Jr., 1972). The streams in this region primarily run through oak or pine woodland.

The spring-dominated streams in southwest Oregon and northern California are located along the border of the

**Table 1.** Summary of data collected for spring-fed and runoff-fed streams. Elevation, GPS, bankfull stream width, and wood length were collected from © Google Earth Pro. Streams marked with * were included in histogram analysis, and those marked with an "a" were used to examine whether wood placement changed over time. Stream width, wood length, and mean discharge are reported as mean±SD when statistics are available. (1) Whiting and Moog (2001), (2) Hygelund (2002), (3) USGS (2018), (4) Deas (2006), (5) Griffiths et al. (2008), (6) Maramec Spring Park (2018), (7) Arguez et al. (2010), (8) Kull and Grosjean (2000), (9) Munoz-Saez et al. (2018), (10) Manga (1996), (11) Wilkerson Jr. (2003), (12) Jefferson et al. (2010), and (13) Vandike (1996). Bankfull discharge values attributed to 3* are estimated as the 1.25-year flood from US Geological Survey (USGS) data. Values marked as "n/a" are not applicable.

| Stream | Elevation (m) | Lat/long (degrees) | Stream width (m) | Wood length (m) | Mean discharge (m$^3$ s$^{-1}$) | Bankfull discharge (m$^3$ s$^{-1}$) | Watershed area (km$^2$) |
|---|---|---|---|---|---|---|---|
| Oregon Cascades: average temperature[7]: 8–10 °C, mean annual precipitation[7]: 0.3–1.3 m, mean annual snowfall[7]: 0.5–0.7 m, land use: pine woodland, grassland, wetland, small farms | | | | | | | |
| Spring-fed | | | | | | | |
| 1  Blue Springs, OR* | 1273 | [42.69580, −122.07173] | 4.3±1.3 | 6.3±1.7 | | 0.089[1] | 0.48[1] |
| 2  Browns Creek, OR | 1334 | [43.72212, −121.80372] | 15.4±2.1 | 16.0±3.2 | | 1.22[1] | 55.9[1] |
| 3  Cultus River, OR*,a | 1357 | [43.88801, −121.76216] | 30.0±3.0 | 17.6±4.0 | 1.8±0.6[3] | | 42.7[3] |
| 4  Deschutes River, OR*,a | 1358 | [43.81417, −121.77583] | 11.1±2.7 | 13.1±3.6 | 4.2±2.1[3] | | 342[3] |
| 5  Fall River, OR*,a | 1276 | [43.79367, −121.52416] | 15.4±6.6 | 16.1±3.4 | | 4.39[2] | 117[10] |
| 6  Lost Creek, OR[a] | 520 | [44.17542, −122.05447] | 16.7±5.2 | 18.22±5.8 | 5.91[12] | | 197[12] |
| 7  Quinn River, OR[a] | 1354 | [43.78417, −121.8351] | 17.9±5.2 | 13.6±4.6 | 0.67[10] | 1.06[3*] | undetermined |
| 8  Reservation Spring, OR*,a | 1274 | [42.69984, −121.96478] | 19.7±3.2 | 22.2±5.5 | | 1.58[1] | 0.12[1] |
| 9  Snow Creek, OR*,a | 1274 | [43.87347, −121.76910] | 16.3±3.2 | 14.0±2.0 | | 1.82[1] | 3.58[1] |
| 10  Spring Creek A, OR*,a | 1281 | [42.67034, −121.88592] | 36.1±11.8 | 18.0±3.4 | | 2.01[1] | 72.8[1] |
| 11  Spring Creek B, OR | 1282 | [42.65413, −121.88043] | 41.5±3.8 | 16.8±2.6 | | 6.77[1] | 33.8[1] |
| Runoff-fed | | | | | | | |
| 12  Boulder Creek, OR*,a | 521 | [43.30361, −122.52917] | 15.9±3.5 | 11.4±4.1 | 2.98±4.32[3] | 31.4[3*] | 78.7[3] |
| 13  Crystal Castle Cr C, OR | 1393 | | 0.96[1] | unknown | | 0.0491[1] | 8.95[1] |
| 14  Cultus Creek, OR*,a | 1399 | [43.82273, −121.82770] | 6.9±2.8 | 16.4±4.6 | 0.62±0.84[3] | 3.02[3*] | 86[3] |
| 15  Deer Creek, OR* | 1383 | [43.80461, −121.83833] | 4.3±0.8 | 10.4±2.3 | 0.2±0.3[3] | 0.463[1] | 55.7[3] |
| 16  Hills Creek, OR | 494 | [43.68056, −122.36944] | 15.9±2.3 | 12.0±1.6 | 4.30±5.84[3] | 35.4[3*] | 136.5[3] |
| 17  Little Deschutes River, OR | 1278 | [43.68917, −121.50167] | 12.5±1.7 | 11.4±4.8 | 5.83±4.56[3] | 10.6[3*] | 2225[3] |
| 18  South Fork McKenzie River, OR | 521 | [44.04722, −122.21667] | 19.7±2.3 | 17.7±5.0 | 17.93±16.64[3] | 104.2[3*] | 414[3] |
| Ozarks: average temperature: 2–15 °C, mean annual precipitation: 0.5–1.2 m, mean annual snowfall: 0.2 m, land use[11]: oak/pine woodland | | | | | | | |
| Spring-fed | | | | | | | |
| 19  Big Springs, MO | 131 | [36.95000, −90.99000] | 88.0±18.0 | n/a | 12.8±4.7[3] | | undetermined |
| 20  Maramec Spring, MO | 239 | [37.95000, −91.53000] | 22.1±3.1 | n/a | | 0.044[6] | 803[13] |
| 21  Tucker Bay Spring, MO | 119 | [36.76576, −90.93988] | 17.0±2.4 | 14.3±4.2 | | 37.75[11] | undetermined |
| Runoff-fed | | | | | | | |
| 22  Bourbeuse River, MO[a] | 245 | [38.14692, −91.58089] | 16.9±5.4 | 20.8±1.9 | 4.01±16.79[3] | 249.8[3*] | 350[3] |
| 23  Current River, MO | 272 | [37.44833, −91.67111] | 14.4±3.7 | 17.8±3.4 | 3.75±4.88[3] | 5.8[3*] | 152[3] |
| 24  Huzzah Creek, MO*,a | 203 | [37.97472, −91.20444] | 23.1±3.6 | 14.3±4.6 | 8.08±21.34[3] | 101.9[3*] | 671[3] |
| 25  Little Piney Creek, MO*,a | 211 | [37.90953, −91.90333] | 17.4±4.4 | 18.3±2.2 | 4.74±11.76[3] | 90.9[3*] | 518[3] |
| 26  Meramec River, MO[a] | 208 | [37.99847, −91.36094] | 35.9±9.2 | 24.1±7.7 | 17.13±42.33[3] | 240.4[3*] | 2023[3] |

| Stream | Elevation (m) | Lat/long (degrees) | Stream width (m) | Wood length (m) | Mean discharge (m³ s⁻¹) | Bankfull discharge (m³ s⁻¹) | Watershed area (km²) |
|---|---|---|---|---|---|---|---|
| Eastern Idaho: average temperature[7]: 1–9 °C, mean annual precipitation[7]: 0.2–0.6 m, mean annual snowfall[7]: 0.7–1.6 m, land use[11]: oak/pine woodland, farm | | | | | | | |
| Spring-fed | | | | | | | |
| 27 Big Springs, ID*,a | 1947 | [44.49892, −111.25711] | 58.4±8.9 | 12.5±3.3 | | 20.5[1] | 0.15[1] |
| 28 Billingsley Creek, ID | 913 | [42.81976, −114.87065] | 11.3±1.5 | n/a | | | undetermined |
| 29 Black Sands Creek, MT*,a | 2023 | [44.66017, −111.16191] | 28.0±7.4 | 15.8±1.9 | | 0.7[1] | 0.082[1] |
| 30 Blue Heart Springs, ID | 879 | [42.71034, −114.83000] | 24.8±3.5 | n/a | | 3.1[11] | 0.01 |
| 31 Buffalo River, ID*,a | 1938 | [44.43844, −111.26001] | 14.2±1.8 | 11.4±4.1 | | 0.21[1] | 0.8[1] |
| 32 Chick Creek, ID* | 1935 | [44.42597, −111.21480] | 4.5±1.7 | 11.2±2.7 | | 1.08[1] | 22.9[1] |
| 33 Elk Springs Creek, ID | 1977 | [44.49468, −111.40109] | 1.4±0.4 | 6.7±2.7 | | 0.024[1] | 0.28[1] |
| 34 Lucky Dog Creek A, ID^a | 1951 | [44.48591, −111.26705] | 7.2±0.6 | 11.1±2.5 | | 0.92[1] | 0.15[1] |
| 35 Lucky Dog Creek B, ID* | 1947 | [44.48822, −111.29158] | 6.9±0.7 | 12.5±3.1 | | 1.35[1] | 5.75[1] |
| 36 Mill Creek, ID | 1939 | [44.46311, −111.42967] | 2.7±0.7 | 7.2±1.2 | | 0.19[1] | 1.88[1] |
| 37 Silver Creek, ID | 1478 | [43.32336, −114.10835] | 20.9±1.8 | n/a | 4.0±1.4[3] | 181[1] | |
| 38 Toms Creek A, ID* | 1932 | [44.41647, −111.29339] | 4.3±0.6 | 9.7±3.3 | | 0.0872[1] | 0.94[1] |
| 39 Toms Creek D, ID | 1914 | [44.40137, −111.36421] | 6.2±1.3 | 9.0±1.7 | | 1.18[1] | 14.4[1] |
| 40 Tyler Creek, ID | 2051 | [44.50973, −111.39774] | 1.2±0.3 | 8.1±1.9 | | 0.2[1] | 3.15[1] |
| Runoff-fed | | | | | | | |
| 41 Fall River, ID*,a | 1643 | [44.05611, −111.35861] | 40.1±4.6 | 15.4±3.4 | 23.78±20.39[3] | 82.4[3]* | 873[3] |
| 42 Henry's Fork, ID*,a | 1602 | [44.11611, −111.33056] | 62.2±6.9 | 23.7±2.8 | 28.14±11.95[3] | 54.4[3]* | 1699[3] |
| 43 Moose Creek, ID* | 1950 | [44.48355, −111.28622] | 2.3±0.3 | 9.9±4.3 | | 0.64[1] | 39.7[1] |
| 44 Robinson Creek, ID | 1606 | [44.11444, −111.32417] | 14.5±2.9 | 11.4±3.4 | 3.59±3.80[3] | 13.1[3]* | 334[3] |
| El Tatio geyser basin, Chile: average temperature[8]: 3.6 °C, mean annual precipitation[8]: 0.0025[8] m, land use: desert, geyser basin | | | | | | | |
| Spring-fed | | | | | | | |
| 45 Río Salado, Chile | 4300 | [−22.33903, −68.01808] | 8.53[9] | n/a | | 0.86[9] | undetermined |
| 46 Stream 0, Chile | 4300 | [−22.33444, −68.03292] | 3.0[9] | n/a | | 0.25[9] | undetermined |
| Northern California: average temperature[7]: 10–12 °C, mean annual precipitation[7]: 1.2–1.6 m, mean annual snowfall[7]: 0.1–1.3 m, land use: oak/pine woodland, shrubland, grassland, farm | | | | | | | |
| Spring-fed | | | | | | | |
| 47 Big Springs Creek, CA | 789 | [41.60115, −122.42650] | 38.2±8.3 | n/a | | 1.74 | undetermined |
| 48 Hat Creek, CA | 1321 | [40.68911, −121.42278] | 7.6±2.0 | 9.9±2.5 | 4.0±1.3[3] | 42[3] | 421[3] |
| 49 Lost Creek, CA | 886 | [39.57003, −121.16534] | 8.7±1.4 | 8.9±3.4 | | 39.7[1] | undetermined |
| Runoff-fed | | | | | | | |
| 50 McCloud River, CA* | 335 | [41.11083, −122.09534] | 28.3±7.2 | 10.3±3.1 | 29.6±41.4[3] | 28.0[3]* | 1564[3] |

| Stream | Elevation (m) | Lat/long (degrees) | Stream width (m) | Wood length (m) | Mean discharge (m³ s⁻¹) | Bankfull discharge (m³ s⁻¹) | Watershed area (km²) |
|---|---|---|---|---|---|---|---|
| Mogollon Rim, Arizona: average temperature[7]: 17 °C, mean annual precipitation[7]: 0.8 m, mean annual snowfall[7]: 0.9 m, land use[5]: oak/pine woodland, wetland meadow | | | | | | | |
| *Spring-fed* | | | | | | | |
| 51 Unnamed Spring 1, AZ | 2207 | [34.47111, −111.28761] | 0.4[5] | n/a | | $1.1 \times 10^{-2}$[5] | 0.029[5] |
| 52 Unnamed Spring 2, AZ | 2313 | [34.43378, −111.16097] | 0.16[5] | n/a | | $2.2 \times 10^{-3}$[5] | 0.025[5] |
| 53 Unnamed Spring 3, AZ | 2313 | [34.43528, −111.16036] | 0.22[5] | n/a | | $2.7 \times 10^{-3}$[5] | 0.0077[5] |
| 54 West Pinchot Spring, AZ | 2146 | [34.50228, −111.19647] | 0.29[5] | n/a | | $3.4 \times 10^{-3}$[5] | 0.011[5] |
| 55 Whistling Spring, AZ | 2289 | [34.44844, −111.19028] | 0.29[5] | n/a | | $2.6 \times 10^{-3}$[5] | 0.028[5] |
| *Runoff-fed* | | | | | | | |
| 56 Buck Springs Canyon, AZ*,a | 2286 | [34.43972, −111.13972] | 1.5 ± 0.5 | 1.5 ± 0.5 | | $6.1 \times 10^{-3}$[5] | 0.84[5] |
| 57 Merritt Draw, AZ | 2291 | [34.44889, −111.19014] | 0.9 ± 0.4 | 8.4 ± 3.8 | | $6.1 \times 10^{-3}$[5] | 0.51[5] |
| 58 Quaking Aspen Canyon, AZ | 2267 | [34.43919, −111.33889] | 0.6[5] | unknown | | $6.1 \times 10^{-3}$[5] | 0.94[5] |

Cascade–Sierra Mountains and the Basin and Range physiographic provinces (Fenneman, 1931). This area lies in the rain shadow of the Cascades to the west. Mean annual precipitation, dominated by snow, decreases from over 1 m to the west to about 0.5 m in the southern part of the study area (Arguez et al., 2010), and mean annual temperatures range from 8 to 12 °C (Arguez et al., 2010). The area is underlain by Quaternary basalt and basaltic andesite. Typical land cover for the studied streams in this region are oak or pine woodland, grassland, shrubland, wetland, and some small farms.

The streams studied in northern Arizona are located along the Mogollon Rim (Pierce et al., 1979). The high relief of the Mogollon Rim at 2100 m induces a strong orographic effect (NRCS, 2005), yielding some of the highest precipitation in the state, an annual average of more than 800 mm (Arguez et al., 2010), and the mean annual temperature is 17 °C (Arguez et al., 2010). The area is underlain by Tertiary basalts, Permian limestone (Kaibab Formation), and sandstone (Coconino sandstone), with streambed material made up of valley fill alluvium (Moore et al., 1960). Watersheds included in this study run through oak/pine woodland and wetland meadows.

The streams in the Ozarks are located in the Potosi, Eminence Gasconade, and Roubidoux formations (Panfil and Jacobson, 2001). The area is underlain by carbonate with interbedded chert and sandstone (Panfil and Jacobson, 2001). Mean annual temperatures range from 2 to 15 °C, and precipitation is 0.5–1.2 m yr⁻¹ (Arguez et al., 2010).

The streams in El Tatio geyser basin, Chile, are located on the San Pedro Formation (Harrington, 1961). Located in the Atacama Desert, precipitation is very low at 0.025 m yr⁻¹, but the high elevation means that the mean annual temperature is 3.6 °C (Kull and Grosjean, 2000). This area is underlain by andesites, dacites, and rhyolites (Harrington, 1961), with the streambed material consisting of glacial outwash. The streams in this area run through desert landscapes above treeline. These streams are included for comparison between spring-fed streams with and without wood since these streams are above treeline and have no recent history of LW. Other spring-fed streams with no visible LW in this study may have had LW in recent history since the watersheds they run through contain forests.

Spring-fed streams occur in specifically defined geological settings in which a highly permeable material overlays an impermeable layer, such as in the volcanic regions explored in this study (Whiting and Stamm, 1995). The geologic setting is important for producing the conditions for spring-fed streams to exist and sustain. Due to these particular geological constraints, it is difficult to find a large, comparable set of runoff-fed streams. We select a set of streams that are located as closely as possible to the spring-fed streams in this study to control for geology as much as possible. We can verify that the labeled runoff-fed and spring-fed streams display different hydrograph behavior by examining the mean and standard deviation of flow, when available. All spring-

fed streams with available data exhibit standard deviations in discharge smaller than their mean discharge, whereas the runoff-fed streams show standard deviations larger than their mean. When unavailable, we rely on the cited authors to correctly identify the flow source for the stream.

## 3 Methods

High-resolution satellite imagery has been shown to be effective in capturing quantitative data about stream morphology and LW (e.g., Leckie et al., 2005; Senter et al., 2017). Using © Google Earth Pro high-resolution imagery (generally 0.15 m resolution – high enough resolution to get accurate measurements, as suggested by Ruiz-Villanueva et al., 2016), we measured stream width along 10 stream cross sections along a reach including the GPS point in Table 1 for 38 spring-fed and 20 runoff-fed streams. This study was limited to exploring width as opposed to width-to-depth ratio because the remote sensing data collection cannot document channel depth. Spring-fed and runoff-fed streams were distinguished based on prior identification in research publications. The GPS points are located at or near the gauges cited. These measurements were compared to field measurements by Whiting and Moog (2001) and Hygelund (2002) for validation. By visual inspection of high-resolution satellite imagery, we determined whether a stream contains wood. Those with no visible wood and those without clear enough imagery are excluded from analyses about wood. In 2018, multiple attempts were made to contact managers of each spring-fed stream where no wood was observed, but we did not receive any responses.

For 25 spring-fed and 19 runoff-fed streams containing wood, we measured the length of 10 or more pieces of LW found in or near the channel in this same reach (Table 1). Additional measurements were taken for streams exhibiting a high degree of variability in wood length. This measurement is meant to characterize the wood source to the streams, so wood found near the streams should be representative of the wood that enters the channel. If wood were only measured in the channel, then the results may be biased since we only measured wood for which we could confidently identify both ends. In the channel, this criterion ruled out many pieces of wood, often excluding smaller pieces or pieces where one end is obscured by trees. Wood outside the channel was sometimes more clearly identifiable in aerial imagery. To verify the validity of this technique, we compared field measurements of wood length at one site to results from remotely sensed measurements. While fully submerged logs likely have an impact on stream morphology as well, they were largely not included in this study due to unreliable identification via satellite imagery. For the remainder of the paper, the term "wood length" refers to the average wood length.

To test the precision of our technique of measuring length in © Google Earth Pro, we measured the length of a single log 10 times in a row to yield a length of $17.6 \pm 0.2$ m with 90 % confidence. The small size of the confidence interval (1.2 %) suggested relatively high precision for the technique. All LW observed via satellite imagery and in the field at this location was long, so no estimates on accuracy of the method for measuring small pieces of LW were possible.

For streams marked by a * in Table 1, we also took histograms of log orientation for single logs in each stream. Histograms were taken using © Google Earth Pro imagery. Ideally, we could measure wood orientation on a scale from 0° (directly in line with flow) to 180° (directly opposite to flow). This is possible in the field, but due to limitations in imagery resolution, we were unable to reliably distinguish the bottom and top of LW in this study. As a result, we noted orientation of LW on a scale from 0° (parallel to flow) to 90° (perpendicular to flow), unable to note orientation (±90°).

More detailed geomorphic and sedimentological data were collected by Whiting and Moog (2001), Hygelund (2002), and Griffiths et al. (2008). Discharge data reported were separated between bankfull and mean discharge in Table 1 for clarity, although for spring-fed streams, since discharge is fairly constant, bankfull discharge and mean discharge are nearly the same (e.g., Whiting and Stamm, 1995; Whiting and Moog, 2001; Manga, 1996). For streams with adequately clear satellite imagery, histograms of wood orientation were made by using © Google Earth Pro to measure the angle between wood orientation and the adjacent stream bank for all wood outside of logjams (approximately 100 pieces) in a stream segment containing the GPS coordinate in Table 1. The reaches varied in length depending on the ease of identifying single logs from about 1.5 km for Cultus River, OR, to over 30 km for McCloud River, CA, although most reaches used for this analysis were under 10 km. We additionally observed, for streams with multiple dates of clear imagery, whether there was any detectable change in wood placement for at least 20 observed logs between dates. Dates were typically from about 2005 to about 2018 with variation in the specific years and time periods when imagery was available. Regional precipitation records did not indicate persistent drought through the entire time period at any site (Arguez et al., 2010), although local conditions may deviate from regional averages. We primarily observed single pieces of LW with few or no logjams in the studied spring-fed streams. We quantified this observation by measuring the density of single logs and the density of logjams over a reach about 500 m in length for streams with adequately clear imagery. These data also allowed for a sense of how close LW is to one another. This is important since the effect of LW on bank erosion is increased when single logs are close together (Zhang, Rutherfurd, and Marren, 2019). We found all best fit parameters using the Levenberg–Marquardt algorithm.

Discharge data were obtained from a range of sources. When available, mean and standard deviation were reported.

For spring-fed streams, mean discharge was similar to bank-full discharge (e.g., Whiting and Stamm, 1995; Whiting and Moog, 2001; Manga, 1996), so when bankfull discharge was not available, mean discharge was used for analyses. For runoff-fed streams, if bankfull discharge was unavailable, the 1.25-year return period was used as an estimate for bank-full discharge. Statistics were repeated with and without estimated bankfull discharge.

Data were modeled to determine which physical factors are most statistically related to stream width. We began from the historical convention of $w = aQ^b$, where $w$ is width, $Q$ discharge, and $a$ and $b$ are constants, which were fit separately for each model and data set. Additional tested models incorporated wood length $l$ in a few different ways. The proposed models we tested were

1. $w = aQ^b$;

2. $w = al^c$;

3. $w = alQ^b$;

4. $w = lQ^b$;

5. $w = al^cQ^b$,

where $w$ is stream width, $l$ wood length, $Q$ discharge, and $a$–$c$ are constants. Models 3 and 4 appear nearly the same, but we fit them separately since model 4 requires fewer fit parameters. These formulae align with the body of research that confirms a power law relationship between stream width and discharge, while taking into account a power law or linear relationship between wood length and stream width for spring-fed streams. We assessed the value of candidate models using adjusted $R^2$ (Miles, 2014), which accounts for the number of predictive variables included in the model, and Akaike's information criterion (AIC), which measures the amount of information lost when data are approximated by a given model as compared to other candidate models also accounting for the number of predictive variables (Akaike, 1974). An adjustment for small sample sizes (AICc) was presented by Hurvich and Tsai (1989), which we used in this study. If the set of AICc values is $\{AICc_i\}$, then the probability that model $i$ is the best of a set of candidate models is given by $e^{(\min(\{AICc_i\})-AICc_i)/2}$.

## 4   Results

### 4.1   Wood dynamics

We begin with a description of the observed wood dynamics within the studied streams. In order for single logs to drive changes in channel morphology, we assume that logs must be immobile in the channel. In order to confirm that this is the case in spring-fed, but not runoff-fed, streams, we examine histograms of wood orientation.

In order to examine the validity of orientation data taken remotely, we compare our orientation results to those of Hygelund (2002) for Cultus River and Cultus Creek, shown in Fig. 1. These sites were chosen from the data available in Hygelund (2002) due to their close proximity to one another and differing flow regime. A visual representation of the differences between the two streams is shown in Fig. 3a for Cultus River, OR (spring-fed), and Fig. 3b for Cultus Creek, OR (runoff-fed), which both feed into Crane Prairie Reservoir. As shown in Table 1, the measured lengths of LW at both streams are about 17 m long. The mean discharge of Cultus Creek from 1923 to 1991 was $0.55 \, \mathrm{m^3 \, s^{-1}}$, with the 95th percentile of flow $q_{95} = 2.3 \, \mathrm{m^3 \, s^{-1}}$, while mean discharge in the Cultus River was $1.5 \, \mathrm{m^3 \, s^{-1}}$, with $q_{95} = 2.8 \, \mathrm{m^3 \, s^{-1}}$ (USGS, 2018). Despite the similar peak flows, Cultus River ($30.0 \pm 3.0$ m) is nearly 5 times wider than Cultus Creek ($6.9 \pm 2.8$ m). In Fig. 3b, there are also numerous large logjams visible in Cultus Creek, whereas very few are visible in Cultus River (Fig. 3a), and those present are small. This comparison is representative of the types of reaches found in spring-fed versus runoff-fed streams included in this study.

Using a Kolmogorov–Smirnov two-sample test, we find that for the measurements in Cultus River (Fig. 1a), there is an 80 % chance that the measurements are from the same distribution and a 15 % chance for the measurements on Cultus Creek (Fig. 1b). The latter low confidence could be due to the fact that the measurements were taken in different years and possibly in different stream segments, and we argue that the qualitative behavior of the histograms is similar enough to draw the same conclusions about wood orientation. Generally, we find that there is relatively good agreement, at least qualitatively, between the in-field results obtained by Hygelund (2002) and those we obtained via satellite imagery.

Following Hygelund (2002), we note that from the histogram of aggregated data for spring-fed streams in Fig. 2a, it appears that wood is preferentially oriented around 50–90° (see the Supplement for individual stream histograms). If wood were mobile in streams, we would expect to see preferential orientation at 0–20° (Braudrick and Grant, 2000). We compare the histogram for spring-fed streams to that for runoff-fed streams in this study, where wood is preferentially oriented around 0–20°. While the aggregate histograms exhibit clear results, many individual histograms demonstrate differences from these trends (see the Supplement). We considered whether basin size impacted the results since larger basins tend to transport more wood (Ruiz-Villanueva et al., 2016), but that observation does not explain the data differences. For instance, Chick Creek, ID (a spring-fed stream), contains wood mostly oriented around 0 or 50°, while Moose Creek, Deer Creek, and Buck Springs Canyon (runoff-fed streams) show random orientation, and Boulder Creek (runoff-fed) is preferentially oriented around 30–50°. In Chick Creek, LW is significantly longer than the width of the stream, so the flow regime in the channel may have little impact on the orientation of wood. In the runoff-fed streams, the deviations from the trend are likely due to other

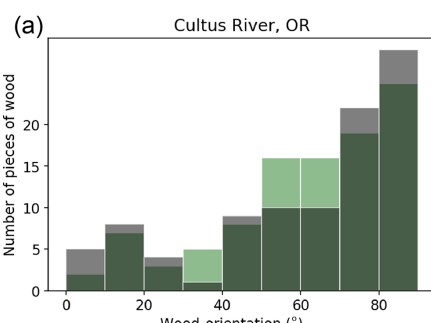
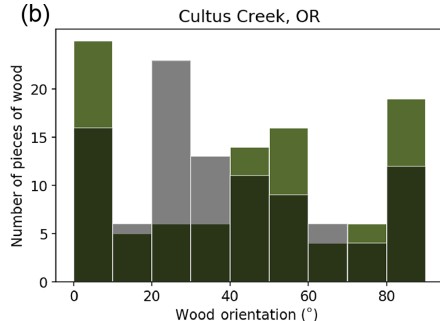

**Figure 1.** Orientation of wood was measured from an adjacent bank for approximately 100 pieces of wood using © Google Earth Pro (green). Hygelund (2002) measured orientation of wood in the field (transparent black). Data are shown together for **(a)** Cultus River and **(b)** Cultus Creek. The distributions align very well for Cultus River and have the same qualitative shape for Cultus Creek, although the center peak is displaced between the two sets of measurements.

aspects of wood dynamics noted during data collection. First, most wood observed in runoff-fed streams was found in log-jams, and identifying single logs to measure the orientation was difficult. In runoff-fed streams in this study, there were on average 37 pieces of single wood per kilometer as opposed to the 130 pieces of single wood per kilometer found in spring-fed streams, as shown in Fig. 4. The high density of single logs means that LW is closely spaced in the streams in this study. This disparity also prevented us from collecting as much data in certain streams due to a dearth of single logs. We noticed about five logjams per kilometer in runoff-fed streams compared to about 1 per kilometer in spring-fed streams. This indicates that there may be a bias toward new wood when measuring single pieces in some runoff-fed channels since older wood may be moved to logjams already. This also led to more difficulty in measuring orientation of single logs in some runoff-fed channels when multi-threaded channels made determining orientation with flow more difficult.

We verify conclusions about residence of LW by examining imagery from multiple dates on the streams marked by an *a* in Table 1. Imagery was clear for a period of 3–10 years, depending on the site, and we examined at least 20 pieces of LW at each site. In each spring-fed stream, we were unable to detect any changes in wood placement at any site. In all of the runoff-fed streams except for Buck Springs Canyon, AZ, we observed a change in orientation or location for at least one observed piece of LW. We suggest that no large runoff events occurred during the 3-year period for which clear imagery is available at Buck Springs Canyon. We thus confirm that there is little mobility of wood in the spring-fed streams in this study, distinct from the motion observed in runoff-fed streams.

### 4.2 Discharge and width

A common relationship used to describe stream width is the Leopold power law relating width $w$ and discharge $Q$ by constants $a$ and $b$ (Leopold and Maddock, 1953): $w = aQ^b$.

Typically, the value of $b$ is close to 0.5, but $b$ can vary depending on the streams being analyzed (Gleason, 2015). Whiting and Moog (2001) found $b = 0.57$ for the spring-fed streams in their study. The finding of Whiting and Moog (2001) suggests that discharge impacts the width of streams in their study to a similar degree as for most channels. We verify the result of Whiting and Moog (2001) for the streams in their study by finding $b = 0.55 \pm 0.1$.

For the full set of spring-fed streams in this study containing wood, we find that [TS1] $a = 9.9 \pm 1.2$ and $b = 0.42 \pm 0.09$ with a Pearson correlation coefficient of 0.52. Spring-fed streams without wood are fit by a statistically different trend-line given by $a = 14.4 \pm 1.4$ and $b = 0.67 \pm 0.08$ with a Pearson correlation coefficient of 0.87. Runoff-fed streams are significantly different from spring-fed streams containing wood only in the coefficient $a$, with $a = 5.1 \pm 1.1$ and $b = 0.36 \pm 0.03$ with a Pearson correlation coefficient of 0.89 (when repeated without estimated bankfull discharges, the results are statistically indistinguishable except for an increase in $R^2$ to 0.99). The value of $a$ is significantly smaller for the runoff-fed streams than the spring-fed streams included in this study. This corresponds to much narrower widths for the runoff-fed streams, confirming the results of Whiting and Moog (2001). It is noteworthy that the correlation coefficient for spring-fed streams with wood is much lower than that for the other two groups, indicating that there is another very important factor needed to describe width adequately.

### 4.3 LW and width

We compare the stream widths we measured to those measured by Whiting and Moog (2001) for the subset of streams included in both studies. For all of the streams contained in both studies, the widths measured by Whiting and Moog (2001) fall within the confidence interval for the widths measured in this study via remote sensing.

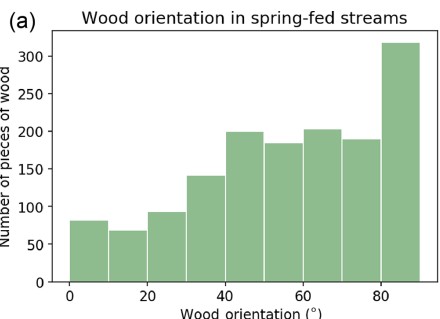
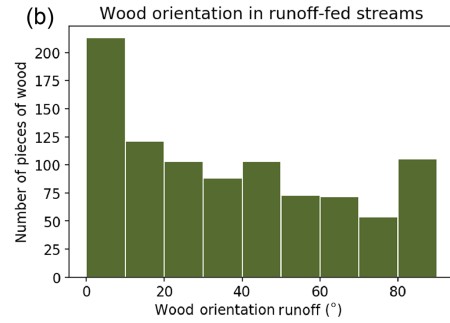

**Figure 2.** Using © Google Earth Pro, orientation of wood was measured from an adjacent bank for approximately 100 pieces of wood in each stream which had clear enough imagery to reliably identify LW (marked by a * in Table 1). Histogram data are aggregated for **(a)** spring-fed and **(b)** runoff-fed streams. Wood in spring-fed streams is preferentially oriented 50–90°, whereas wood in runoff-red streams is more randomly oriented with a significant portion of wood oriented 0–20°.

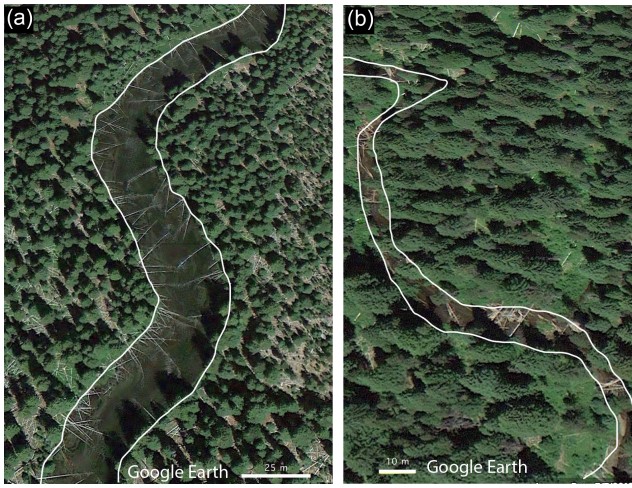

**Figure 3.** © Google Earth Pro high-resolution imagery showing **(a)** Cultus River ($q_{95} = 2.8 \, \mathrm{m^3 \, s^{-1}}$) and **(b)** Cultus Creek ($q_{95} = 2.3 \, \mathrm{m^3 \, s^{-1}}$). Stream channels are outlined in white, and flow direction is down from the top of the image in both panels. These images are representative of the general wood dynamics in the two streams, where most of the wood in panel **(a)** is single logs, and most of the wood in panel **(b)** is in logjams, so little of the wood in panel **(b)** would contribute to the histogram shown in Fig. 2b.

We additionally compare field measurements of wood length of 10 pieces of LW at Cultus River, OR [43.82381, −121.79687], to remotely sensed wood length data for 10 pieces of LW at the same location. In the field, we find that the wood length was $18.5 \pm 5.0 \, \mathrm{m}$, and via remote sensing, we measured $17.4 \pm 3.9 \, \mathrm{m}$. We note that the standard deviation in wood length is larger for field measurements than satellite measurements. This indicates that satellite measurements likely cause us to both miss short LW (as described above) and underestimate the length of long LW, likely due to obscured ends. However, the confidence intervals for these measurements overlap, so we conclude that it is still accurate to measure wood length via © Google Earth high-resolution satellite imagery.

For the 25 spring-fed streams containing wood, we find that there is a power law relationship between LW length and stream width, as shown in Fig. 6b, with a Pearson correlation coefficient of 0.66. For streams lying below the dashed width-to-length line in Fig. 6a, wood found in and around the streams is typically longer than the streams are wide, while streams above the dashed line are wider than the LW found in the system. Most streams in the study are clustered near the dashed line, so wood length is comparable to stream width. There is variation in the length of LW between streams. This variation is generally geographically explicable, with streams located near one another having similar LW sizes. Also note that in Fig. 6a, the standard deviation for wood length generally increases with increasing stream width. We speculate that larger streams may contain wood that has traveled further and thus exhibits larger variation in size, but we do not have data to confirm this hypothesis. Runoff-fed streams are marked in Fig. 6 by black dots.

The relationship between LW length and stream width is displayed on a $\ln - \ln$ plot in Fig. 6b with the line of best fit for $w = al^b$, where $w$ is stream width, $l$ is wood length, and $a$ and $b$ are constants. The 95 % confidence interval is shaded for $a = 0.04 \pm 0.03$ and $b = 2.4 \pm 0.4$. The Pearson correlation coefficient for this relationship is 0.66, indicating that wood is as strongly correlated to the width of spring-fed streams as is discharge. We see from Fig. 6b that the fit parameters encompass well the variability in the data. The best fit for the runoff-fed streams is not significantly different from that for the spring-fed streams, with a Pearson correlation coefficient of 0.56.

## 4.4 Using LW and discharge to describe stream width

There are large Pearson correlation coefficients for the relationships between wood and width as well as discharge and width for spring-fed streams, implying both are important de-

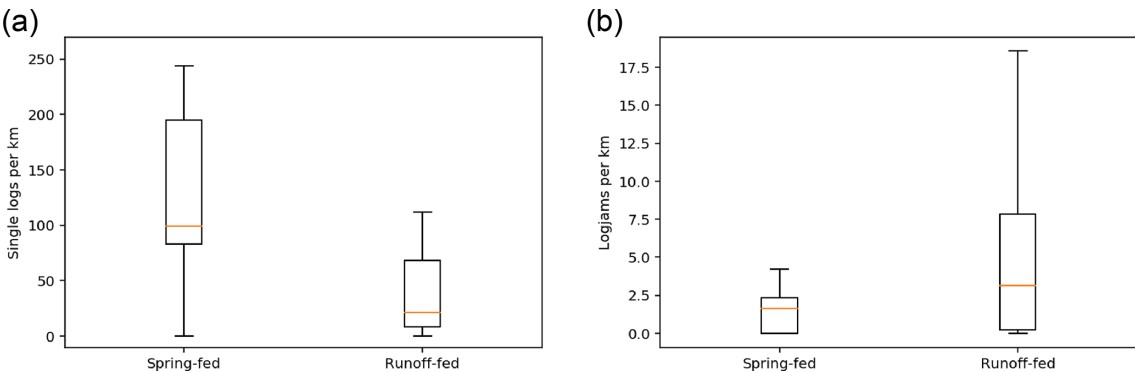

**Figure 4.** Boxplot representing the number of **(a)** single logs and **(b)** logjams identified per kilometer via satellite imagery on spring-fed and runoff-fed streams.

**Table 2.** Fit statistics for candidate models for spring-fed and runoff-fed streams. Adjusted $R^2$ and Akaike's information criterion (AICc) account for the number of predictive variables. A larger $R^2$ value indicates better fit, while a smaller AICc value indicates that less information is lost. The AICc probability is the likelihood that a given model is the best model based on the criterion of lost information as measured by AICc. The results from adjusted $R^2$ match very well with the AICc results in ranking. For both runoff-fed and spring-fed streams, we note that models 3–5 are essentially identical when fit for all streams since parameters $a$ and $c$ in model 5 are indistinguishable from those in model 1.

|   | Function | $a$ | $b$ | $c$ | Adjusted $R^2$ | AICc | AICc probability | $a$ | $b$ | $c$ | Adjusted $R^2$ | AICc | AICc probability |
|---|---|---|---|---|---|---|---|---|---|---|---|---|---|
|   |   |   |   |   | All spring-fed |   |   |   |   |   | Spring-fed $\leq 30$ m |   |   |
| 1 | $w = aQ^b$ | 14.00 | 0.27 |   | 0.25 | 118.0 | 0.36 | 11.65 | 0.16 |   | 0.16 | 84.7 | 0.00 |
| 2 | $w = al^c$ | 0.45 |   | 1.39 | 0.29 | 118.9 | 0.23 | 0.24 |   | 1.53 | 0.62 | 68.0 | 0.99 |
| 3 | $w = alQ^b$ | 1.07 | 0.22 |   | 0.39 | 116.0 | 0.99 | 0.97 | 0.06 |   | 0.54 | 73.1 | 0.08 |
| 4 | $w = lQ^b$ |   | 0.25 |   | 0.44 | 116.2 | 0.91 | 0.05 |   | 0.54 | 0.54 | 73.1 | 0.08 |
| 5 | $w = al^c Q^b$ | 0.93 | 0.22 | 1.06 | 0.39 | 117.0 | 0.60 | 0.24 | 0.00 | 1.53 | 0.60 | 71.0 | 0.22 |
|   |   |   |   |   | All runoff-fed |   |   |   |   |   | Runoff-fed $\leq 30$ m |   |   |
| 1 | $w = aQ^b$ | 9.53 | 0.23 |   | 0.39 | 89.8 | 0.39 | 8.10 | 0.19 |   | 0.67 | 47.2 | 0.11 |
| 2 | $w = al^c$ | 0.19 |   | 1.68 | 0.45 | 87.9 | 0.99 | 1.71 |   | 0.80 | 0.24 | 60.9 | 0.0 |
| 3 | $w = alQ^b$ | 0.92 | 0.11 |   | 0.44 | 89.1 | 0.55 | 0.72 | 0.10 |   | 0.28 | 60.0 | 0.0 |
| 4 | $w = lQ^b$ |   | 0.09 |   | 0.43 | 89.2 | 0.53 |   | 0.03 |   | 0.16 | 61.1 | 0.0 |
| 5 | $w = al^c Q^b$ | 0.62 | 0.09 | 1.16 | 0.44 | 90.0 | 0.34 | 90.96 | 0.34 | −1.11 | 0.78 | 43.0 | 0.99 |

scriptive factors for stream width. There is, however, a ln − ln correlation between discharge and wood length with Pearson correlation coefficient of 0.44, indicating that the two parameters do not contain totally unique information but do contain a significant amount of unique information. Since discharge and wood length are both significant descriptors for stream width and contain unique information, we examine a model for stream width incorporating both parameters. Full results for all tested models are shown in Table 2. For all cases, model ranking is very similar for AICc and adjusted $R^2$.

For all spring-fed streams, model fittings of parameter $a$ in model 3 and $a$ and $c$ in model 5 are indistinguishable from those in model 1, making models 3, 4, and 5 nearly identical, so we discuss only models 1, 2, and 4. Model 4 performs significantly better than models 1 and 2, as demonstrated by a high adjusted $R^2$ and a low AICc value in Table 2. There

is still a significant probability that model 1 or 2 could be the most effective model (36 % and 23 %, respectively), though, since models 1 and 2 resemble model 5 very closely. For spring-fed streams with an average width less than 30 m (the group of streams which are close to or narrower than available LW), models 3 and 4 are indistinguishable and models 2 and 5 are indistinguishable, so we discuss only models 1, 2, and 3. Model 1 (based only on discharge) drops in significance from an adjusted $R^2$ of 0.25 to 0.16, while all other models rise in significance, most notably model 2, which rises from an adjusted $R^2$ of 0.29 to 0.62. This trend is preserved in AICc values, which indicate that model 2 (based only on LW length) is the highest-performing model for spring-fed streams narrower than 30 m.

The fit for all of the proposed models is plotted in graphs for spring-fed (a) and runoff-fed (b) streams in Fig. S3 TS2.

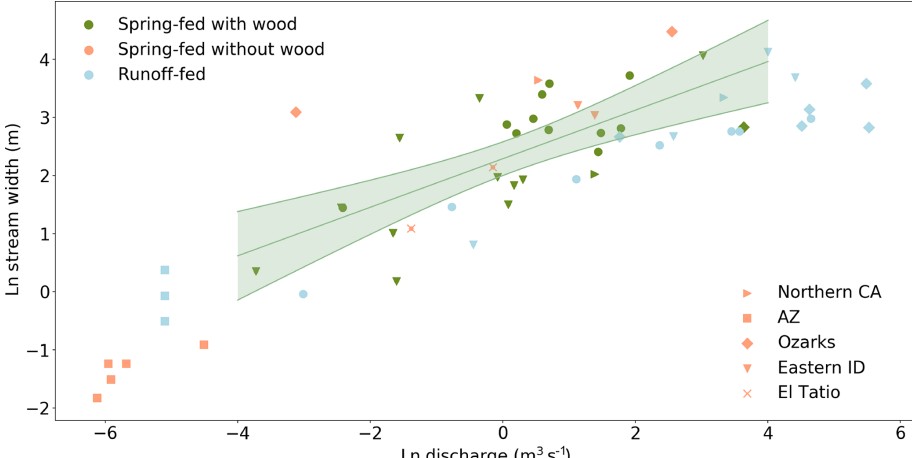

**Figure 5.** Relationship between bankfull discharge (or 1.25-year flow as an approximation to bankfull discharge for some runoff-fed streams marked in Table 1) and stream width plotted on a $\ln - \ln$ plot for spring-fed streams with wood (dark green), spring-fed streams without wood (orange), and runoff-fed streams (light blue). The line of best fit for spring-fed streams containing wood is shown ($w = aQ^b$, $b = 0.42 \pm 0.09$, $a = 9.9 \pm 1.2$); 95 % confidence interval for the fit is shaded. Stream types are denoted by color, as shown in the top left, and locations are denoted by shape, as shown in the bottom right. Runoff-fed streams are fit by a statistically significant different value of $a = 5.1 \pm 1.1$, indicating that runoff-fed streams are narrower than spring-fed streams at the same bankfull discharge. All runoff-fed streams contain wood, and no runoff-fed streams without wood were available for comparison.

## 5 Discussion

### 5.1 Wood dynamics

We found that there is a significant difference between the residence location and residence time of LW in spring-fed and runoff-fed streams. This difference is demonstrated by the different frequencies of single logs versus logjams in runoff-fed and spring-fed streams as well as the orientation histograms for spring-fed and runoff-fed streams. The orientation histogram and historical satellite imagery for spring-fed streams indicate immobile wood, while the histogram and historical satellite imagery for runoff-fed streams indicate frequent log mobility. While it may be more complicated to interpret orientation data in small streams (Kramer and Wohl, 2016), the historical satellite imagery confirms the conclusion that LW is stable in spring-fed streams and often mobile in runoff-fed streams in this study. Even so, wood in larger spring-fed streams is likely more mobile than wood in smaller spring-fed streams since the mean discharge is higher, although mobility in runoff-fed streams appears to be much greater. We also note that the standard deviation in wood length generally increases with increasing stream width in spring-fed streams, while standard deviation in wood length in runoff-fed streams is generally comparable with the standard deviation for larger spring-fed streams in the same geographic region in this study, supporting the hypothesis that increased wood mobility increases the standard deviation in wood length. The clear differences in wood dynamics suggest a different impact of wood on morphology

of spring-fed and runoff-fed streams, in which the impact of single logs may be dominant in the former.

In particular, we note that the wood dynamics observed in spring-fed streams in this study differ from the logjams that would be typically expected for streams in which wood length is similar to or smaller than channel width (Kramer and Wohl, 2016). The preponderance of single logs matches better with the category of small streams, where stream width is less than wood length (Kramer and Wohl, 2016). This difference suggests that adding a criterion for hydrograph variability may be useful in classifying streams impacted by LW. Such a criterion may allow for the classification of spring-fed streams as small due to their low peak discharge relative to the mean.

### 5.2 Discharge and width

Figure 5 shows a distinction between spring-fed streams with and without wood in the relationship between discharge and width. There is, however, only a small set of data points available to identify the relationship for spring-fed streams without wood, and 5 of the 12 streams in this group are unusually narrow for the study group. The remaining points are not visually distinct from the point cloud for spring-fed streams with wood. For the streams in the Ozarks and eastern Idaho, we speculate that these streams may once have had significant amounts of wood due to their size, location in wooded areas, and a history of "management" that may have included wood removal (Willis et al., 2017; Schaper, 2001; Maramec Spring Park, 2018; Silver Creek, 2006). If this is the case, then the presence of wood may have had a lasting impact on

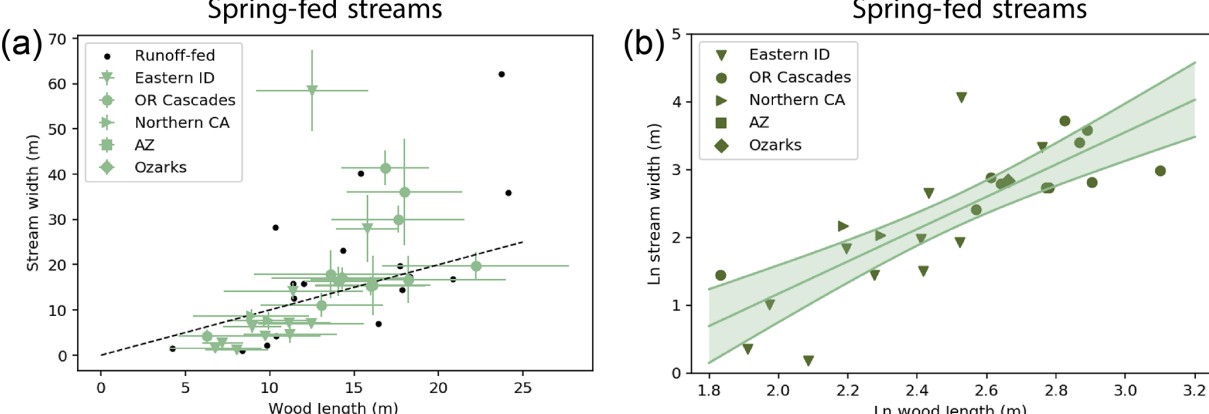

**Figure 6.** Wood length and stream width were measured using © Google Earth Pro satellite imagery. The relationship between wood length and stream width for spring-fed streams is shown **(a)** on a plot with width equal to length shown as a dashed line and error bars showing the standard deviation and runoff-fed streams marked with black dots (error bars left off for clarity of viewing) and **(b)** on a $\ln-\ln$ plot with the line of best fit ($w = al^b$ with $b = 2.4 \pm 0.4$ and $a = 0.04 \pm 0.03$), error bars and runoff-fed streams left out for clarity. The 95 % confidence interval for the line of best fit is shaded. In both panels, the data symbols represent the geographic locations of the streams. There is no apparent significant clustering by location. In panel **(a)**, streams that fall above the dotted line are wider than the wood load entering the streams, whereas the streams falling below the line are narrower than the wood load.

the channel morphology that is still measurable despite the present lack of wood, explaining why those streams lie in the point cloud for streams containing wood. While many, if not all, streams in the study may have been subject to wood removal at some point, we take the current wood load as representative of the type of wood dynamics that would have existed prior to wood removal. Additional management is not expected to have had much impact on results since geomorphic restoration efforts are typically not attempted over large reaches such as those used in this study (Boyer et al., 2003).

In contrast to the US streams, the El Tatio streams are above the treeline so they would not have had wood in the past. It is possible that the channels were shaped by a different hydrological regime, but the streams run through glacial outwash, so the shape of the channel is dynamic and is probably controlled by the contemporary, spring-fed fluvial regime. Including all spring-fed streams in calculating the relationship between stream width and discharge does not significantly change the relationship parameters. This finding indicates that we are unable to reliably distinguish between spring-fed streams with wood and those without, an analysis which may be confounded by the minimal availability of spring-fed streams without wood for data collection.

There is, however, a robust distinction between spring-fed and runoff-fed streams in terms of the relationship between discharge and stream width, demonstrated in the fitted parameter $a$. This parameter indicates that for streams larger than those measured by Griffiths et al. (2008), it is generally the case that spring-fed streams are wider than runoff-fed streams.

### 5.3   LW and width

We expect wood to be most important for describing the width of streams when it is comparable in size to the streams. When wood is much longer than the width of the stream, then additional increases in wood length do not change the way wood interacts with the channel since the majority of the wood piece is outside of the channel since nearly all wood observed in spring-fed streams is oriented closer to perpendicular to the bank than parallel, causing wood to either span the channel or interact with the channel only for part of the LW length. Dixon and Sear (2014) note that LW longer than 2.5 times the channel width is generally immobile. While LW is immobile, though, the full length of the LW is relatively unimportant for its impact on stream width beyond the fact that it is longer than the channel is wide. Conversely, when the stream is much wider than the wood, LW can only be close to the bank on at most one side of the stream. Zhang, Rutherfurd, and Marren (2015) found that when LW at a given orientation is closer to the bank, the impact on shear stress is greater. Taking distance from the bank as the most important predictor of how important a single log is in altering channel properties, decreasing the size of LW after a certain point then does not change the ability of the wood to be close only to one bank. Thus, we expect LW to be less important in two cases: (1) where streams are very narrow in comparison to LW length and (2) where streams are very wide in comparison to LW length. In other words, when discharge is outside a certain range, we expect the impact of LW on stream width to decrease since channels are either very wide in comparison to wood length or very narrow. We see visually in Fig. 6a that when streams are wider than about

25 m, the points deviate significantly from the otherwise apparently linear trend. For streams in this study wider than 30 m, streams are much wider than the LW found in or near them. In fact, we find that there is a linear relationship with a Pearson correlation coefficient of 0.75 for streams smaller than 30 m wide, more significant than the $\ln - \ln$ relationship for all data. This stronger correlation aligns well with our hypotheses about when wood should have an impact on stream morphology, i.e., when LW is comparable in length to stream width. While we are unable to say with confidence whether or not there is a difference between spring-fed streams with or without wood, we find that deviation from the relationship occurs where expected if wood were driving the relationship.

In the case of runoff-fed streams, although the best fit matches closely with that for spring-fed streams, we find it likely that this relationship does not hold in general for runoff-fed streams. Since there is a strong bias in our set of runoff-fed streams toward high-discharge streams, with over 70 % of the runoff-fed streams exhibiting a bankfull discharge higher than $5\,\mathrm{m^3\,s^{-1}}$ and most over $50\,\mathrm{m^3\,s^{-1}}$, it may be a coincidence that the runoff-fed streams included in this study are about as wide as the wood found in them. The difficulty in identifying runoff-fed streams in geologic settings in which spring-fed streams occur prevents us from assessing more fully the relationship between wood length and stream width in runoff-fed streams in a comparable geologic setting.

## 5.4 Using LW and discharge to describe stream width

For all runoff-fed streams with available discharge and wood length data, models 3, 4, and 5 are indistinguishable, so we evaluate only models 1, 2, and 4. The highest-performing model is model 2 (based only on LW length), although models 1 and especially 4 receive high AICc probabilities (39 % and 53 %, respectively). When we restrict analysis to runoff-fed streams narrower than 30 m wide, the adjusted $R^2$ for models 2, 3, and 4 drops significantly, while significance of models 1 and 5 increases. For model 5, though, the fit parameter ($c = -1.1 \pm 0.4$) is negative, completely opposite from that for all runoff-fed streams ($c = 1.2 \pm 0.6$). Due to the small sample size and unexpected sign, we find it unlikely that this model is appropriate in general. If we remove model 5 from consideration, then model 1 is clearly the best remaining model. (When repeated without estimated discharges, $R^2$ values were 0.98 for all models, likely due to the small number of points, allowing for overfitting. The values of $a$ and $c$ are still indistinguishable from those in model 1.)

The large Pearson correlation coefficients for the relationships in spring-fed streams between discharge and width as well as wood length and width indicate that combining both pieces of data into a single model could provide increases in model performance. This initial thought is borne out by the increase in adjusted $R^2$ and decrease in AICc for the model $w = l Q^b$ compared to the relationships for either

wood length or discharge alone. However, when the analysis is repeated for streams narrower than 30 m (where wood is close to the width of the channel), the most significant relationship becomes $w = a l^b$, depending only on wood length. Streams narrower than 30 m are examined separately since this is the group of streams that we hypothesize should be most impacted by LW.

For runoff-fed streams, we repeat the same analyses, and we find no improvement in model performance by including both variables ($Q$ and $l$). Unlike for the case of spring-fed streams, when we again restrict the streams included to those narrower than 30 m, the significance of the relationship between wood length and stream width (model 2) drops significantly, making the relationship $w = a Q^b$ the most significant of the tested relationships. This result agrees with our hypothesis that the good fit between wood length and width is coincidental since removing streams where wood should be less important causes the significance to fall instead of rise. Thus, we conclude that model 2 is likely not the best model for the case of all runoff-fed streams. The next best candidate is model 4, although model 1 is nearly as effective. This suggests that discharge is also the more important model factor for all runoff-fed streams, not just those smaller than 30 m.

The finding that model 4 performs well for both spring-fed and runoff-fed streams is particularly interesting since the form ($w = l Q^b$) resembles the Leopold and Maddock formula except with $l$ instead of $a$. Thus we can think of wood length $l$ as a useful factor in understanding the variations of the coefficient $a$ in different locations.

## 6 Conclusions

We are able to use high-resolution satellite imagery to reproduce measurements taken in the field by Whiting and Moog (2001) and Hygelund (2002), and new measurements taken for this article. It is particularly notable that there is a significant overlap in confidence intervals for the wood lengths measured via remote sensing and in the field. This contribution increases confidence in the use of remote sensing to assess LW accurately and quantitatively. Remote sensing tools provide a more straightforward way to effectively collect data at a large number of field sites.

We verify the result of Whiting and Moog (2001) that spring-fed streams are generally wider than their runoff-fed counterparts. We also identify differences in dynamics of LW between spring-fed and runoff-fed streams which underline the importance of peak flow and flow variability when identifying stream dynamics in relation to LW load. While we are unable to isolate LW as the cause of the difference in morphology between spring-fed and runoff-fed streams, we note that a model for stream width in spring-fed streams based solely on wood length $l$ is the best model tested in this study for streams comparable in size to LW. We therefore recommend further study into mechanisms by which LW may

control the width of spring-fed streams. This result provides deeper insight into what controls the width of streams in general by demonstrating a strong relationship between wood length and stream width when discharge is controlled.

**Data availability.** Datasets related to this article can be found at https://github.com/lapidesd/Lapides_Manga_2019 (Lapides and Manga, 2019).

**Supplement.** The supplement related to this article is available online at: https://doi.org/10.5194/esurf-8-1-2020-supplement.

**Author contributions.** DAL performed data collection and analysis and interpretation of results. MM suggested the project idea and provided substantive feedback and ideas for analysis methods and interpretation.

**Competing interests.** The authors declare that they have no conflict of interest.

**Financial support.** This research has been supported by the Hellman Foundation and the University of California, Berkeley. TS3

**Review statement.** This paper was edited by Jens Turowski and reviewed by two anonymous referees.

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

TS5 Since the link is no longer working, do you have an alternative source?