# Peer review of "Large wood as a confounding factor in interpreting the width of spring-fed streams"

_Earth Surface Dynamics, 2019_

## Referee Comment (RC1) · Anonymous Referee #1 · 14 Nov 2019

The authors have addressed a comparison of the influence of large wood in the channel morphology of spring-fed streams vs runoff-fed streams. This issue has been very little studied, and the authors have highlighted its importance.

In general, they have produced a very interesting manuscript.

I have several observations, some more relevant than others, and I will list all of them following the text.

I consider that some of my observationem are important and need to be solved, so my recommendation is to accept with Major Revision this manuscript. I am not asking for additional analyses, but parts of the manuscript must be revised and completed. I am

willing to revise a new version of this article.

Abstract: 1. Page 1, line 2. The authors write: "Due to the distinctive damped hydrograph of spring-fed streams, large woody debris is less mobile in spring-fed than runoff-fed stream channels". As the authors are introducing the reader to the topic, I suggest completing the phrase with a few words about the distinctive hydrograph of runoff-fed streams; this can appear a little obvious, but I consider it will complete the picture. 2. Page 1, phrase between lines 4 to 6, "We used high-resolution satellite imagery . . . . . . 38 spring-fed and 20 runoff-fed streams". This statement does not fully agree to what is written between lines 60-64, so please revise and rewrite. 3. Page 1, line 8. Additionally, to what? Please revise and complete. 4. Page 1, phrase between lines 6 to 9: what about wood jams? 5. Page, the last phrase of Abstract is very close to a repetition of the previous one. I suggest rewriting and merging them in just one phrase.

Introduction: 6. Page 2, line 26: environmental variables, such as? Please revise and complete. 7. Page 2, phrase beginning in line 30, about the Griffiths et al (2008) publication. The authors write that the study in Arizona was of spring-fed streams, but this phrase ends mentioning a comparison to runoff-fed streams. Please revise and rewrite accordingly. 8. Page 2, phrases between lines 33 to 42. The authors introduced the issue of LWD, and the differences of LWD load between streams. I consider that information of LWD recruitment mechanisms and sources and characteristics of riparian forests in these streams is needed.

Field area: 9. Page 3, line 68. Not clear what the spring-fed streams in El Tatio Geyser Field in Chile are really providing to this research. These streams do not have LWD (see page 5, line 96). This needs extra explanation, here, and in the rest of the manuscript.

Methods: 10. Page 4, line 108. Please complete giving the dimensions of the LWD. 11. Page 4, line 108. Please complete giving the resolution of the high-resolution

imagery. dimensions of the LWD. 12. Page 6, line 128. The precision of the technique of measuring length in Google Earth Pro was tested for longer LWD pieces, but what for shorter pieces? Please complete.

Discussion: 13. Page 14, line 285, the authors mention wood loading. Please explain and complete because this issue is not addressed or described in previous chapters (Methods, Results). 14. Page 15, line 327, the authors seem to consider that most longer wood pieces are outside of the channel. I am not convinced at all with this asseveration; if outside, are these long logs spanning the channel? Please revise and explain. 15. Page 16, line 332. LWD can be less important where streams are very narrow and where streams are very wide, but this is always in relation to wood dimensions (length in this case). Please complete. 16. Page 16, lines 334-336 and then 352 and 356, the authors discuss about streams narrower than 25 and 30 m. Why this difference? Please explain.

---

## Referee Comment (RC2) · Anonymous Referee #2 · 27 Nov 2019

Overall Summary: This article makes observations and comparisons in stream width and wood loading between spring fed streams versus run-off streams of similar discharges. The topic is interesting and the authors provide some compelling data gathered from remote sensing to broaden the traditional perspective that stream width is mostly a function of discharge to also viewing wood as a primary driver of stream widths. The paper is well written and the statistics well done. I originally reviewed this article for another journal and supplied a fairly critical review with the suggestion to reject based on some very major concerns. I was pleasantly surprised to find that this version of the manuscript is substantially altered from the original submission and thoroughly addressed most of my concerns from the original review. It is very much

improved and is now ready for publication. I suggest accept with minor edits.

Minor Edits/Concerns: The presence of wood jams versus single logs plays a role in not just channel width, but impacts whether channels are multi-thread versus single thread. Often jams are a forcing mechanism for multi-thread channels. I would have like to see this pointed out/ discussed a little bit within the introduction and in context of the case study streams. Were all the streams studied single channel streams? Or did the run-off streams have multi-threads along with the log jams? I don't expect this to become a major point of the paper, but I do think it is salient when interpreting the width of streams due to wood loading. Example locations to include this- ex. Lines 55-58. Wood accumulations may not just increase width or width to depth ratios, especially for places where they promote avulsions and multi-threads (see 10.1525/bio.2013.63.6.6, 10.1177/0309133314548091)

Line 53: This hypothesis could have a citation to back it up

It was unclear to me which study streams had erodible beds versus which ones didn't. Some descriptions mentioned that the stream went through erodible materials (i.e. glacial outwash, alluvium, etc) but other areas just mentioned that the underlying hardrock geology. It is important to know if all the streams in this study had erodible banks versus streams in bedrock channels. Can all the streams adjust their planform to the flow and to wood? Perhaps table one can include a field that specifies whether banks were erodible or not.

Caption for figure 4 should mention that these two streams have similar flow.

Results presented in line 237 should be highlighted more in the conclusion etc. As yet there really isn't that much in the wood literature looking at the accuracy of measuring wood from aerial photographs or satellite imagery. This is a valuable contribution and it would be nice to see it a little more light shone on it, rather than having it be buried in the middle of the paper.

Symbol shapes in legend in Figure 5 don't match symbols in plot

The findings regarding differences in mobility between spring fed and run-off could be highlighted more strongly in the discussion. They were presented in the results and thus should be discussed more fully in the discussion. For example, I found the increase in std deviation of width with stream width to be an interesting finding... but the speculation that this has to do with mobility and wood travelling further a little bit contradictory with your finding that the wider spring fed streams have less mobility than the narrower run-off streams. Some more discussion about mobility differences is warranted.

Personally I prefer the acronym LW rather than LWD. I don't like the negative connotation of 'debris' assigned to LW in streams. I would encourage the authors to consider not using LWD (keep it as a key word for searchability)

The last sentence of the conclusion seems out of place. I would delete it. Since the paper never really goes into management, it seems out of place to mention it out of nowhere in the last sentence of the paper.

---

## Editor Comment (EC1) · Jens Turowski (Editor) · 28 Nov 2019

Dear authors,

we have received two reviews now, and both referees are largely positive about the paper. There are numerous requests for clarification and more details on the data, and suggestions for additional discussion and for improving language. I do not see the need to elaborate on anything; the reviewers' comment seem to be clear and detailed. Overall, the necessary revisions seem to be moderate to me.

Please submit a revised manuscript with a detailed rebuttal letter.

[Figure]

I am looking forward to seeing the revisions,

Best wishes, Jens Turowski
* * *

---

## Author Comment (AC2) · 4 Dec 2019

Thank you for your comments. We have altered the manuscript to address the suggestions and concerns that were raised. This review is thorough and thoughtful, and much appreciated. Specific changes are described below.

Thank you for pointing out the importance of logjams for causing channels to develop multiple threads. There are small reaches in some of the streams in this study that are multi-threaded, and these reaches contain a significant amount of wood and logjams. We included this observation in the manuscript and added a brief statement about the impact of logjams on multi- or single-threads in channels around Line 55, as

recommended.

We added a citation for logjams in runoff-fed streams in the hypothesis on Line 53, as recommended.

For clarity, we included a statement at the beginning of the field area section that notes that all streams in the study run through erodible material. The underlying hardrock geology is required to produce the upwelling of flow for the spring-fed channels, but the channels themselves are able to adapt quickly.

We added q_95 values to the caption for Figure 4 to demonstrate that the streams have similar flow.

It is nice to hear that quantifying the accuracy of satellite-derived wood measurements is so valuable. We added an extra couple of sentences in the conclusion to highlight that the comparison between field measurements and remote sensing yielded good agreement, increasing confidence in the accuracy of remote sensing for producing quantitative results.

We added symbol shapes to a legend in Figure 5 so that both colors and shapes are clearly labeled in the figure.

We added more discussion about wood mobility, specifically focused on the finding of increased std of wood length with increasing stream width in spring-fed streams. The std for wood length in runoff-fed streams is generally comparable with the std for wood length in larger spring-fed streams in the same geographic area. Although mobility appears to be higher in runoff-fed streams than spring-fed streams, the increase in std with increasing stream width in spring-fed streams may be indicative of increased mobility compared to smaller spring-fed streams. There is likely a maximum std given the population of wood available in a geographic area.

We appreciate the suggestion to replace the acronym LWD with LW to avoid the negative connotations associated with 'debris.' We now use the acronym LW.

We removed the last sentence of the conclusion.

**ESurfD**

---

## Author Response (AR1)

**Referee #1:**

Summary: The authors have addressed a comparison of the influence of large wood in the channel morphology of spring-fed streams vs runoff-fed streams. This issue has been very little studied, and the authors have highlighted its importance. In general, they have produced a very interesting manuscript. I have several observations, some more relevant than others, and I will list all of them following the text. I consider that some of my observationem are important and need to be solved, so my recommendation is to accept with Major Revision this manuscript. I am not asking for additional analyses, but parts of the manuscript must be revised and completed. I am willing to revise a new version of this article.

Abstract: 1. Page 1, line 2. The authors write: "Due to the distinctive damped hydrograph of spring-fed streams, large woody debris is less mobile in spring-fed than runoff-fed stream channels". As the authors are introducing the reader to the topic, I suggest completing the phrase with a few words about the distinctive hydrograph of runoff-fed streams; this can appear a little obvious, but I consider it will complete the picture. 2. Page 1, phrase between lines 4 to 6, "We used high-resolution satellite imagery . . . . . . 38 spring-fed and 20 runoff-fed streams". This statement does not fully agree to what is written between lines 60-64, so please revise and rewrite. 3. Page 1, line 8. Additionally, to what? Please revise and complete. 4. Page 1, phrase between lines 6 to 9: what about wood jams? 5. Page, the last phrase of Abstract is very close to a repetition of the previous one. I suggest rewriting and merging them in just one phrase.

Introduction: 6. Page 2, line 26: environmental variables, such as? Please revise and complete. 7. Page 2, phrase beginning in line 30, about the Griffiths et al (2008) publication. The authors write that the study in Arizona was of spring-fed streams, but this phrase ends mentioning a comparison to runoff-fed streams. Please revise and rewrite accordingly. 8. Page 2, phrases between lines 33 to 42. The authors introduced the issue of LWD, and the differences of LWD load between streams. I consider that information of LWD recruitment mechanisms and sources and characteristics of riparian forests in these streams is needed.

Field area: 9. Page 3, line 68. Not clear what the spring-fed streams in El Tatio Geyser Field in Chile are really providing to this research. These streams do not have LWD (see page 5, line 96). This needs extra explanation, here, and in the rest of the manuscript.

Methods: 10. Page 4, line 108. Please complete giving the dimensions of the LWD. 11. Page 4, line 108. Please complete giving the resolution of the high-resolution imagery. dimensions of the LWD. 12. Page 6, line 128. The precision of the technique of measuring length in Google Earth Pro was tested for longer LWD pieces, but what for shorter pieces? Please complete.

Discussion: 13. Page 14, line 285, the authors mention wood loading. Please explain and complete because this issue is not addressed or described in previous chapters (Methods, Results). 14. Page 15, line 327, the authors seem to consider that most longer wood pieces are outside of the channel. I am not convinced at all with this asseveration; if outside, are these

long logs spanning the channel? Please revise and explain. 15. Page 16, line 332. LWD can be less important where streams are very narrow and where streams are very wide, but this is always in relation to wood dimensions (length in this case). Please complete. 16. Page 16, lines 334-336 and then 352 and 356, the authors discuss about streams narrower than 25 and 30 m. Why this difference? Please explain.

**Response to Referee #1:**

Summary response to review: Thank you for your comments. We have altered the manuscript to address the concerns that were raised. The bulk of the comments address the scaling between channel dimensions and LWD size. This perspective is valuable and greatly appreciated. Specific changes are detailed below.

Response to comments on abstract: We have altered the abstract to 1. add a statement about the hydrograph of runoff streams, 2. Include the full content of lines 60-64, 3. Correct language for clarity, 4. Add a statement about logjams, 5. Remove repetitive phrases.

Response to comments on introduction: We have altered the introduction to 6. Clarify the meaning of the sentence, 7. Clarify the intent of the Griffiths et al (2008) study, 8. Describe recruitment mechanisms.

Response to comments on Field area: We have altered the field area section to 9. Describe why the El Tatio streams were included in this study. They are the most reliable example of spring-fed streams not containing wood. In order to assess whether wood causes spring-fed streams to be wide, we would like to compare spring-fed streams with wood to spring-fed streams without wood. The only other spring-fed streams in this study that don't contain wood likely had wood in the past since they run through forested watersheds.

Response to comments on methods: 10. The table contains stream width and LWD dimensions. We altered the methods section to 11. State the likely resolution of Google Earth Pro imagery and 12. Describe why precision was tested only for longer LWD pieces. All of the LWD observed at the site visited was long. There were no shorter pieces available.

Response to comments on discussion. We altered the discussion to 13. A clearer description of the meaning of the sentence, 14. Explain that we suggest that longer wood is spanning the channel or the majority of a given piece (not the majority of wood pieces) is outside the channel when the channel is comparatively narrow, 15. Clarify that channel width is in comparison to wood dimensions, and 16. Describe that streams wider than 30 m (corrected to 30 in all places) are significantly wider than LWD in this study. Thus, we hypothesize that LWD length should be more important for streams narrower than 30m than those wider than 30m in this study.

**Referee #2:**

Overall Summary: This article makes observations and comparisons in stream width and wood loading between spring fed streams versus run-off streams of similar discharges. The topic is interesting and the authors provide some compelling data gathered from remote sensing to broaden the traditional perspective that stream width is mostly a function of discharge to also viewing wood as a primary driver of stream widths. The paper is well written and the statistics well done. I originally reviewed this article for another journal and supplied a fairly critical review with the suggestion to reject based on some very major concerns. I was pleasantly surprised to find that this version of the manuscript is substantially altered from the original submission and thoroughly addressed most of my concerns from the original review. It is very much improved and is now ready for publication. I suggest accept with minor edits.

Comment 1: The presence of wood jams versus single logs plays a role in not just channel width, but impacts whether channels are multi-thread versus single thread. Often jams are a forcing mechanism for multi-thread channels. I would have like to see this pointed out/ discussed a little bit within the introduction and in context of the case study streams. Were all the streams studied single channel streams? Or did the run-off streams have multi-threads along with the log jams? I don't expect this to become a major point of the paper, but I do think it is salient when interpreting the width of streams due to wood loading. Example locations to include this- ex. Lines 55-58. Wood accumulations may not just increase width or width to depth ratios, especially for places where they promote avulsions and multi-threads (see 10.1525/bio.2013.63.6.6, 10.1177/0309133314548091)

Comment 2: Line 53: This hypothesis could have a citation to back it up

Comment 3: It was unclear to me which study streams had erodible beds versus which ones didn't. Some descriptions mentioned that the stream went through erodible materials (i.e. glacial outwash, alluvium, etc) but other areas just mentioned that the underlying hardrock geology. It is important to know if all the streams in this study had erodible banks versus streams in bedrock channels. Can all the streams adjust their planform to the flow and to wood? Perhaps table one can include a field that specifies whether banks were erodible or not.

Comment 4: Caption for figure 4 should mention that these two streams have similar flow.

Comment 5: Results presented in line 237 should be highlighted more in the conclusion etc. As yet there really isn't that much in the wood literature looking at the accuracy of measuring wood from aerial photographs or satellite imagery. This is a valuable contribution and it would be nice to see it a little more light shone on it, rather than having it be buried in the middle of the paper.

Comment 6: Symbol shapes in legend in Figure 5 don't match symbols in plot

Comment 7: The findings regarding differences in mobility between spring fed and run-off could be highlighted more strongly in the discussion. They were presented in the results and thus should be discussed more fully in the discussion. For example, I found the increase in std deviation of width with stream width to be an interesting finding. . . but the speculation that this has to do with mobility and wood travelling further a little bit contradictory with your finding that the wider spring fed streams have less mobility than the narrower run-off streams. Some more discussion about mobility differences is warranted.

Comment 8: Personally I prefer the acronym LW rather than LWD. I don't like the negative connotation of 'debris' assigned to LW in streams. I would encourage the authors to consider not using LWD (keep it as a key word for searchability)

Comment 9: The last sentence of the conclusion seems out of place. I would delete it. Since the paper never really goes into management, it seems out of place to mention it out of nowhere in the last sentence of the paper.

**Response to Referee #2:**

Response to review summary: Thank you for your comments. We have altered the manuscript to address the sugestions and concerns that were raised. This review is thorough and thoughtful, and much appreciated. Specific changes are described below.

Comment 1: Thank you for pointing out the importance of logjams for causing channels to develop multiple threads. There are small reaches in some of the streams in this study that are multi-threaded, and these reaches contain a significant amount of wood and logjams. We included this observation in the manuscript and added a brief statement about the impact of logjams on multi- or single-threads in channels around Line 55, as recommended.

Comment 2: We added a citation for logjams in runoff-fed streams in the hypothesis on Line 53, as recommended.

Comment 3: For clarity, we included a statement at the beginning of the field area section that notes that all streams in the study run through erodible material. The underlying hardrock geology is required to produce the upwelling of flow for the spring-fed channels, but the channels themselves are able to adapt quickly.

Comment 4: We added $q\_95$ values to the caption for Figure 4 to demonstrate that the streams have similar flow.

Comment 5: It is nice to hear that quantifying the accuracy of satellite-derived wood measurements is so valuable. We added an extra couple of sentences in the conclusion to highlight that the comparison between field measurements and remote sensing yielded good agreement, increasing confidence in the accuracy of remote sensing for producing quantitative results.

Comment 6: We added symbol shapes to a legend in Figure 5 so that both colors and shapes are clearly labeled in the figure.

Comment 7: We added more discussion about wood mobility, specifically focused on the finding of increased std of wood length with increasing stream width in spring-fed streams. The std for wood length in runoff-fed streams is generally comparable with the std for wood length in larger spring-fed streams in the same geographic area. Although mobility appears to be higher in runoff-fed streams than spring-fed streams, the increase in std with increasing stream width in spring-fed streams may be indicative of increased mobility compared to smaller spring-fed streams. There is likely a maximum std given the population of wood available in a geographic area.

Comment 8: We appreciate the suggestion to replace the acronym LWD with LW to avoid the negative connotations associated with 'debris.' We now use the acronym LW.

Comment 9: We removed the last sentence of the conclusion.

[revised manuscript text omitted]

---

## Author Response (AR2)

Associate Editor Decision: Publish subject to technical corrections (14 Jan 2020) by Jens Turowski

Comments to the Author:

Dear authors,

thank you for the revised manuscript. All queries have been addressed and the manuscript is ready for publication. I am impressed by the high quality of writing; reading the paper I have not found a single language problem or typo. However, I do have a small suggestion you may want to consider: the paragraph at the end of page 14, starting on line 293, may be better placed in the discussion, as it goes beyond the mere presentation of results and contains elements of interpretation. Please consider moving it into a suitable location in the discussion. The point is minor and the organisation works even if you decide not to follow this suggestion.

I have decided on technical corrections. Once you upload the revised version, the manuscript will go straight to production, without another editorial screening.

Thanks and best wishes, Jens Turowski

Response from authors:

Thank you so much for your feedback. We decided to move the paragraph at 293 to the Discussion, as suggested. We feel that the review process has been productive, and we are excited to publish this paper.

Comments to the author:
I am happy to confirm acceptance of your paper for publication in esurf (and I apologize for the delay in getting this decision to you). The two reviewers and the Associate Editor (AE) all viewed your submission positively, and it appears that you have diligently and thoughtfully taken on board the suggestions of the reviewers — as well as providing a manuscript in overall very good shape, as the AE notes. I expect your paper will make an important contribution to understanding of channel morphology and of the role for large wood in geomorphic systems.

I have noted a few minor suggestions below. You might consider making these adjustments before uploading the final version of your manuscript, or during proof correction. I also encourage you to think carefully about the AE's suggestion of moving the text starting at line 293, though I leave that decision to you.

Thank you for choosing esurf for your work, and I hope you consider publishing in this venue again in the future!

Josh

Line 38: It seems a little unusual at this point in the Introduction to bring the text to focus on "streams included in this study" rather than remaining general, so you might consider keeping a more general perspective (e.g., "In many settings, including those considered in this study, ..."

Lines 115: standard deviations in what? I assume this is discharge, but might help to specify "standard deviations in discharge smaller than their mean"

Methods section in general: you jump between present and past tense; it would be better to keep this consistent

Line 122: do you mean "including AT the GPS points in Table 1"?

Table 1:
-- "GPS" might be better stated as "lat/long", also should note that the is in degrees
-- the for baneful discharge heading, the m3/s should have superscript 3
-- units for area are presumably km2, not km3

Line 134: criterion, not criteria

Line 154-155: I was curious how long these stream segments typically were?

Line 170: c is not a parameter in the equation w=aQ^b, so probably better to leave this out here and specify at line 176 instead?

Line 176: if you say here at c and b are constants, shouldn't you include a, too?

Figure 1 caption
-- "are have" isn't correct (presumably you mean just "have"?)
-- it might help readers to specify which of these is spring fed and which is runoff fed

Figure 2 caption
-- Google should be capitalized
-- remove "from" before 50-90°?

Line 222: just "imagery were clear"?

Lines 228-235: I was surprised to find this text here, because you had discussed these two systems earlier, at lines 193-200. Is there merit in considering moving the text at 228-235 up so that it brings together the discussion of these two systems? That would also mean switching the order of Figures 3 and 4, which also makes some sense to me.

Line 246: "When" should not be capitalized

Figure 5: this is blurry in the version of the manuscript that I have; please just double check when you receive the proofs that this (and all of the other figures) appear clearly

Line 286: extra space before 36%

Line 287: Do you mean that models 1 and 2 resemble model 4? Otherwise this logic is not completely clear to me, with reference to the prior sentence.

Line 305: what does "time of LW" mean?

Line 332: either make point cloud one word (as at line 328) or two words, as here

Line 337: no comma after El Tatio streams

Thank you for providing the data via GitHub!

Response from the authors:

Thank you for your feedback and for accepting the manuscript for publication. We are excited that this work will appear in esurf. All of the suggestions made are greatly appreciated, and we have made corrections for all of them. We reviewed the manuscript for tense consistency and adjusted the tense where appropriate. Any comments that could have additional explanation are below.

Thank you so much.

Line 154-155: The length of these segments varied based on how much wood was easily visible. For streams with a great deal of wood and clear imagery (like Cultus River), a reach of about 1.5 km was sufficient to collect about 100 data points on wood orientation. For other streams, especially runoff-fed streams, this was more difficult. For some runoff-fed streams, the entire length of the stream was scoured to get as many data points as possible. McCloud River is the most extreme case, with a reach of about 30 km explored to find single logs in the channel.

[revised manuscript text omitted]